# Unsupervised Multitaper Spectral Method for Identifying REM Sleep in Intracranial EEG Recordings Lacking EOG/EMG Data

**DOI:** 10.3390/bioengineering10091009

**Published:** 2023-08-25

**Authors:** Kyle Q. Lepage, Sparsh Jain, Andrew Kvavilashvili, Mark Witcher, Sujith Vijayan

**Affiliations:** 1School of Neuroscience, Sandy Hall, Virginia Tech, 210 Drillfield Drive, Blacksburg, VA 24060, USA; akvavi4@vt.edu (A.K.); neuron99@vt.edu (S.V.); 2Department of Biomedical Engineering and Mechanics, Virginia Tech, 325 Stanger St., Blacksburg, VA 24061, USA; sparsh@vt.edu; 3Section of Neurosurgery, Carilion Clinic, Carilion Roanoke Memorial Hospital, 1906 Belleview Ave SE, Roanoke, VA 24014, USA; mrwitcher@carilionclinic.org

**Keywords:** intracranial, EEG, neural dynamics, oscillations, sleep scoring, spectral analysis, multitaper

## Abstract

A large number of human intracranial EEG (iEEG) recordings have been collected for clinical purposes, in institutions all over the world, but the vast majority of these are unaccompanied by EOG and EMG recordings which are required to separate Wake episodes from REM sleep using accepted methods. In order to make full use of this extremely valuable data, an accurate method of classifying sleep from iEEG recordings alone is required. Existing methods of sleep scoring using only iEEG recordings accurately classify all stages of sleep, with the exception that wake (W) and rapid-eye movement (REM) sleep are not well distinguished. A novel multitaper (Wake vs. REM) alpha-rhythm classifier is developed by generalizing K-means clustering for use with multitaper spectral eigencoefficients. The performance of this unsupervised method is assessed on eight subjects exhibiting normal sleep architecture in a hold-out analysis and is compared against a classical power detector. The proposed multitaper classifier correctly identifies 36±6 min of REM in one night of recorded sleep, while incorrectly labeling less than 10% of all labeled 30 s epochs for all but one subject (human rater reliability is estimated to be near 80%), and outperforms the equivalent statistical-power classical test. Hold-out analysis indicates that when using one night’s worth of data, an accurate generalization of the method on new data is likely. For the purpose of studying sleep, the introduced multitaper alpha-rhythm classifier further paves the way to making available a large quantity of otherwise unusable IEEG data.

## 1. Introduction

Intracranial electroencephalography (iEEG) recordings are routinely collected during the surgical treatment of epilepsy, while simultaneous electromyography (EMG) recordings and electrooculography (EOG) recordings are not [1,2]. Since the typical hospital stay of an epileptic patient undergoing invasive monitoring is 1 to 2 weeks, nearly every epilepsy unit in the US has iEEG sleep data that they have collected, and that they continue to collect, which is unaccompanied by electromyography (EMG) and electrooculography (EOG). These latter two recording modalities measure muscle tone and eye movements, respectively, and are required by expert human sleep scorers to separate wake from rapid eye movement (REM) sleep [3]. Reliable identification of REM in this data would rescue a large quantity of rare data collected during invasive surgery from living human brains. This data could be used to, for example, (i) assess the clinical use of REM dynamics to accurately identify the epileptic zone [4,5,6]; such identification is required to successfully treat focal epilepsy seizures surgically [7,8,9], and (ii) to further our understanding of brain dynamics during REM sleep (e.g., [10]).

Scientists have characterized sleep into four stages, namely: non-REM stage 1 (NREM1), non-REM stage 2 (NREM2), non-REM stage 3 (NREM3), and REM sleep. These stages are based, in part, upon electrophysiological features observed in recordings such as EEG and iEEG. Of all the sleep stages, REM sleep exhibits electrophysiological features that are most similar to those exhibited in recordings collected from awake subjects. As a consequence, the separation of REM from the awake state (hereafter labeled “Wake”), is the most difficult classification task when accurately labeling electrophysiological recordings (e.g., see [11,12]). This labeling process is sometimes referred to as “sleep scoring” by sleep scientists, and individuals sleep scoring data are colloquially known as “sleep scorers”. Since REM sleep is accompanied by low muscle tone and rapid eye movements, expert sleep scorers rely upon accompanying recordings of muscle activity (electromyography, EMG) and upon recordings that indicate eye movement (electrooculography, EOG) to separate Wake from REM sleep [13,14,15,16,17]. As previously stated, these recordings are absent from the majority of iEEG data, and a method capable of identifying REM based upon iEEG data alone will be of significant value.

Sleep scoring is a laborious and time-consuming process. As a consequence, many methods of automatically sleep scoring electrophysiological data exist; examples include refs. [18,19,20,21]. However, they have not been wholly successful, particularly when separating REM from Wake [11,12]. This limitation has not been alleviated with the advent of modern machine learning and artificial intelligence (AI) methods. In a review of 37 published deep learning sleep classifiers spanning the ten years from 2010 to 2020, it is recommended that EOG and EMG be analyzed in conjunction with EEG to achieve robust classification results [22]. More recently, classical machine learning methods have been compared against deep learning methods such as those reported in [23,24,25] for the purpose of scoring sleep and are found to perform similarly [26,27]. It is worth noting, however, that the performance reported in [24] on EEG is excellent, possibly due to the use of time-frequency distributions, possibly due to the use of a convolutional neural network, or possibly due to both.

Regardless, it is important to note that iEEG is not EEG, and it exhibits key differences. First, iEEG is significantly more invasive than EEG, with multi-contact electrodes penetrating through the cortex of the brain to clinically-indicated deep structures such as the hippocampus and the amygdala. With EEG, the electrodes lie upon the scalp with consistent, and more evenly spaced, coverage from individual to individual. Second, iEEG typically lacks occipital recordings. These two differences are substantial, and any sleep-scoring method that classifies sleep well using EEG recordings may not classify sleep well when applied to iEEG sleep data. For intracranial EEG, there is one published method of scoring sleep that does not use commensurate EMG and EOG recordings [28]. This method of sleep scoring also performs least well when identifying REM and N1 sleep stages (see Figure 3, [28]).

Our method uses alpha oscillations recorded intracranially and unaccompanied by EOG or EMG recordings to identify REM sleep episodes. Alpha (8–12 Hz) oscillations occur during eyes-closed wakefulness in humans but dissipate once the subject falls asleep [29,30,31,32]. This pattern of alpha activity has been used for the purpose of sleep scoring EEG data by human sleep scorers, especially to distinguish awake periods from periods of rapid eye movement (REM) sleep [4,33]. The alpha oscillations described above, as detected using standard EEG, are most prominent over the occipital cortex. However, intracranial patients rarely have electrodes in the occipital regions; instead, intracranial electrodes are commonly located in the temporal and frontal regions of the brain. Figure 1 displays an example of the difference in alpha oscillations between Wake and REM sleep exhibited in iEEG.

Unlike the other previously mentioned iEEG sleep scoring method (i.e., [28]), the proposed Wake vs. REM classifier is unsupervised (i.e., there is no training), its goal is to identify REM, it is restricted to a single 4 Hz frequency interval (i.e., alpha rhythm), and it uses K-means clustering modified to cluster multiple-channel, multitaper eigencoefficients.

Similar to the sleep scoring method presented in ref. [28], the classifier proposed here requires only one night’s worth of data. Applying the method introduced in ref. [28] to our data yields hypnograms that agree less with those identified by human sleep scorers than those depicted in Figure 4 in ref. [28]. This is the case despite trying all the software options provided in ref. [35] and selecting the analyzed channels using three different methods. Specifically, (i) selecting all available channels (this is the default “BEA” option in ref. [35]), (ii) selecting only the subset of iEEG channels used to compute the results in this paper, or (iii) selecting those channels that exhibit fewer than one ictal discharge per minute using the recommended ictal event detector [36]. These last two methods use the “EEA” option specified in ref. [35]. Since we have not attained the performance reported in ref. [28], we leave further work directly comparing our method with those introduced in ref. [28] for a future study.

Numerous automatic EEG sleep scoring methods are reported in the literature [37,38]. A typical sleep scoring classifier involves feature selection, often through the use of clustering, followed by sleep stage classifier training. K-means clustering of EEG frequency domain features is reported in refs. [39,40,41,42]. In refs. [39,40], K-means clustering of reduced EEG spectral features is used to obtain feature weights to use with a K-nearest neighbor classifier and a decision tree classifier for the purpose of scoring sleep data. In ref. [42], K-means clustering is applied to the spectrogram of the Cz EEG channel for semi-automated sleep-scoring. In ref. [41], feature vectors containing EEG power spectra evaluated between the frequencies of 0.5 and 20Hz are clustered to demonstrate subject-specific differences. Our method differs from these methods in that (i) its purpose is to identify REM from iEEG recordings, (ii) it is focused upon a single interval of frequencies, (iii) it is completely unsupervised and does not train a classifier, nor (iv) does it reduce the number of features in a feature-reduction step, (v) it uses aspects of the multitaper method of spectral time-series analysis to construct multi-channel alpha rhythm feature vectors, and (vi) the presented K-means clustering algorithm is updated from the original K-means algorithm to make use of the large-sample distribution of the resulting multitaper, multi-channel, alpha rhythm features.

## 2. Methods

### 2.1. Data Collection

Data were collected from nine patients (6 females and 3 males) between the age of 19 and 55 years (mean $33.11\pm 13.99\;\left(SD\right)$ years) undergoing invasive monitoring using intracranial depth electrodes for intractable epilepsy (see Table 1). The study from which data was collected was approved by the Carilion Clinic IRB, IRB-20-1179, 8 February 2021, and was conducted in accordance with the Declaration of Helsinki. Patients signed informed consent prior to participating in the study. Two to sixteen depth electrodes, placed according to clinical criteria (see Table 2), each with 6 to 14 regularly-spaced macro contacts, were implanted in each subject, and the electric potential difference between each contact and the reference contact (a subdermal electrode contact placed near Fz), were sampled at 2kHz. For each subject, recording started between 8 PM and midnight, depending on an estimated bedtime. EEG data from the raw Neuralynx files was converted to EDF files in blocks of 4 h periods. For each subject, between 6 and 12 h of sleep/wake data were recorded and considered for analysis. Electrophysiological readings from Ad-Tech electrodes are collected at 2kHz with a 256-channel ATLAS digital acquisition system, (Neuralynx, Bozeman, MT), which was connected to a PC running Neuralynx, Pegasus Acquisition Software, version 2.1.1.

### 2.2. Electrode Localization

The locations of depth electrode contacts are determined from preoperative high-resolution magnetic resonance imaging (MRI) scans along with postoperative CT scans. For each subject, the Freesurfer program [43,44] is used to generate 3D RAS coordinates and Desikan-Killiany anatomical labels [45] for the electrode contacts, and then the postoperative CT scan is coregistered with these coordinates using the IELU pipeline and a MATLAB script [46,47] to obtain anatomical locations.

### 2.3. Manual Sleep Scoring

For the purposes of comparison, manual sleep scoring is conducted independently by two experienced sleep scorers using criteria specified by the American Association of Sleep Medicine for sleep scoring [4] using synchronized EEG (EEG is collected from a centrally located subdermal strip), iEEG, EMG, and EOG signals. All recordings are collected using the ATLAS recording system. Periods identified as REM sleep are further verified by two additional sleep scorers. An epoch is labeled REM if low-amplitude mixed-frequency activity is observed along with rapid eye movements and a low chin EMG tone. Of the subjects examined, only subject 7 displays unusual electrophysiological phenomena. Though epochs for subject 7 labeled as NREM or REM exhibit many features of their respective sleep stages, all sleep stages also display unusual activity. For instance, during periods of low EMG tone accompanied by conjugate eye movements that would otherwise be labeled REM, high voltage, low frequency activity (but not low enough frequency to be N3) is often observed on some channels. This behavior is not typical for any of the other subjects. Subject 7 is included for completeness and also as an example of how the proposed REM/Wake classifier performs on unusual activity.

### 2.4. Data Selection

Analysis is restricted to at least one seizure-free night of sleep where the subject entered both NREM and REM sleep and to surface contacts on frontal and temporal depth electrodes. These contacts are more likely to be (i) seizure-free, (ii) collected by other iEEG data centers, (iii) may generalize to electrocorticography (ECoG) data, which records activity at the surface level of the brain, and (iv) are most likely to exhibit the neural REM/Wake alpha-rhythm dynamics observed in standard EEG recordings.

Electrode contacts are further removed from analysis if identified by epileptologists to lie in the epileptic focus or to exhibit interictal activity (see Table 1 and Table 2).

### 2.5. Filtering and Sectioning

To suppress power-line interference, selected data is filtered to remove 60 Hz and harmonic frequencies using a zero-phase FIR notch filter. Filtered data is partitioned into 30 s, non-overlapping sections. Each section is assigned a stage label by the manual sleep scoring procedure (see Section 2.3).

### 2.6. Multitaper Eigencoefficients

Each 30 s section of data is further subdivided into 5 non-overlapping sections, each of which is 6 s in duration. Let c∈N equal the electrode-contact index, s∈N indicate the subject, *e* indicate the 30 s episode (or section), $m\in \left\{1,2,3,4,5\right\}$ the subsection, and let *t* equal the subsection time-index. Further define, $d_{m,t}^{\left(c,s,e\right)}$ to be equal to the $t^{th}$ centered sample of the $m^{\mathrm{th}}$ subsection of the $e^{\mathrm{th}}$ 30 s window for contact *c* and subject *s*. Note that the explicit dependence of c,e upon *s* is suppressed in the notation. Let $\nu_{t}^{\left(k\right)}$ be the $t^{\mathrm{th}}$ sample of the $k^{th}$-order discrete-prolate spheroidal sequence (DPSS); a DPSS is parameterized by order *k*, as well as by its length *N* and bandwidth *W*. The zero-th order DPSS possesses, among all length *N* sequences, the maximum signal energy within the frequency interval (−W,W). The $k+1^{\mathrm{st}}$ order DPSS is, of all sequences orthogonal to $\nu^{\left(0\right)}$, maximally energy concentrated within (−W,W) [48]. Here we focus on alpha-rhythm, which is classically considered bandlimited to the frequency interval (8Hz,12Hz). For the remainder of this work *f* are equal to 10Hz and *W* is equal to 2Hz. For sample period Δ, the corresponding dimensionless DPSS time-bandwidth parameter is equal to ΔNW. It is equal to 12 for the chosen ΔN and *W*. Let $F_{N}$, equal to 1/2Δ, be the Nyquist frequency. The corresponding $k^{\mathrm{th}}$ order eigencoefficient, $Y_{e,m,k}$, evaluated at the frequency $f\in -f_{N},f_{N}$, is equal to [49],
(1)$$Y_{m,k}^{\left(c,s,e\right)}\left(f\right)=\sum_{t=0}^{N-1}v_{t}^{\left(k\right)}d_{m,t}^{\left(c,s,e\right)}\;e^{-i2\pi ft\Delta }\;.$$
Here, $i^{2}$ is equal to −1 and $Y_{m,k}^{\left(c,s,e\right)}\left(f\right)\in C$ is a complex-valued number. A simple multitaper spectrum estimate, ${\overset{ˇ}{S}}^{\left(c,s,e\right)}$, evaluated at *f*, is equal to,
(2)$${\overset{ˇ}{S}}^{\left(c,s,e\right)}\left(f\right)=\left.\sum_{m=1}^{5}\sum_{k=1}^{K}{\left|Y_{m,k}^{\left(c,s,e\right)}\left(f\right)\right|}^{2}/5K\right.\;.$$
The parameter *K* is chosen to be the number of DPSSs that possess signal-energy within the frequency interval (−W,W) exceeding 0.98 (K=22). The signal-energy of a sequence is equal to its squared $\ell_{2}$ norm. The $\ell_{2}$ norm of any DPSS is equal to 1. With this specification, out-of-band bias due to spectral leakage is limited by the optimal in-band energy concentration properties of the DPSSs [34,49,50,51]. The resulting *K* is equal to 22 (K=2NW−2). Let $N_{c}$ equal the number of electrode contacts for subject *s*. Form the matrix of eigencoefficients $Y^{\left(s,e\right)}\in C^{N_{c}\times 110}$ characterizing alpha rhythm for the $e^{\mathrm{th}}$ 30-s recording for subject *s*. The $p^{\mathrm{th}}$ row and $q^{\mathrm{th}}$ column of $Y^{\left(s,e\right)}$ is equal to,
(3)$${\left[Y^{\left(s,e\right)}\right]}_{p,q}={\left.Y_{m_{q},k_{q}}^{\left(p,s,e\right)}\left(f\right)\right|}_{f=10\;\mathrm{Hz}}\;,$$
such that $q=22\left(m_{q}-1\right)+k_{q}$ and subject to the restriction $m_{q}\in 1,\ldots ,5$, $k_{q}\in 1,\ldots ,22$.

### 2.7. Multitaper Eigencoefficient Clustering

Cluster labels are assigned in a K-means clustering algorithm modified to cluster the multitaper eigencoefficients characterizing each 30 s recording episode. Specifically, the distance introduced in [52,53] is replaced with the cluster-conditioned probability density, P(Y|C=j), for cluster *j*. Furthermore, the probability density of observing the eigencoefficient feature matrix Y, conditioned upon cluster *j* is,
(4)$$\begin{array}{c} \ln P\left(Y\;|\;C=j\right)=\;-N_{c}\ln\left(2\pi \right)+\\ \;-\frac{N_{c}}{2}\left(\ln\left|{\overset{ˇ}{R}}_{r}^{\left(j\right)}\right|+\ln\left|{\overset{ˇ}{R}}_{i}^{\left(j\right)}\right|\right)-\\ \;\;\frac{1}{2}tr\left\{{\left(Y_{r}-{\overset{ˇ}{\mu }}_{r}^{\left(j\right)}1_{Nr}^{T}\right)}^{T}{\left[R_{r}^{\left(j\right)}\right]}^{-1}\left(Y_{r}-{\overset{ˇ}{\mu }}_{r}^{\left(j\right)}1_{Nr}^{T}\right)\right\}-\\ \;\;\frac{1}{2}tr\left\{{\left(Y_{i}-{\overset{ˇ}{\mu }}_{i}^{\left(j\right)}1_{Nr}^{T}\right)}^{T}{\left[R_{i}^{\left(j\right)}\right]}^{-1}\left(Y_{i}-{\overset{ˇ}{\mu }}_{i}^{\left(j\right)}1_{Nr}^{T}\right)\right\}\;. \end{array}$$
Here, $N_{r}$ is equal to 110. It is equal to the number of subsections (i.e., 5) multiplied by the number of DPSS tapers, *K* (i.e., 22). The real-valued matrices $Y_{r}$ ($Y_{i}$) are the real (imaginary) components of the eigencoefficient feature matrix Y, ${\overset{ˇ}{\mu }}^{\left(j\right)}\in C^{N_{c}}$ is the $j^{\mathrm{th}}$ cluster-conditional sample average, and ${\overset{ˇ}{R}}^{\left(j\right)}\in C^{N_{c}\times N_{c}}$ is the $j^{\mathrm{th}}$ cluster-conditional sample covariance matrix. Let the $j^{\mathrm{th}}$ element of the set of labels $L_{j}\in {\left\{1,\ldots ,N_{cluster}\right\}}^{N_{episode}}$ indicate the cluster to which the eigencoeffiecient feature-vector $Y^{\left(s,e\right)}$ is assigned. Here, $N_{cluster}$ is the number of clusters and $N_{episode}$ is the number of 30 s epochs. For subject *s*. Let $Q_{j}$ be the number of episodes assigned to cluster *j*. Furthermore, for cluster *j*, the cluster-specific mean-vector estimate ${\overset{ˇ}{\mu }}^{\left(j\right)}$ is equal to,
(5)$$\begin{array}{ccc} {\overset{ˇ}{\mu }}^{\left(j\right)} & = & \left.\sum_{e=1}^{Q_{j}}\sum_{j^{\prime }=1}^{N_{r}}{\left[Y^{\left(s,e\right)}\right]}_{.,j^{\prime }}/\left(N_{r}Q_{j}\right)\right.\;. \end{array}$$
Let the convenience variable ${\tilde{Y}}^{\left(s,e,j\right)}$ equal
(6)$$\begin{array}{ccc} {\tilde{Y}}^{\left(s,e,j\right)} & = & Y^{\left(s,e\right)}-{\overset{ˇ}{\mu }}^{\left(j\right)}1_{Nr}^{T}\;. \end{array}$$
Here, $1_{Nr}$ is a column-vector of Nr elements, each of which equals 1. The real part, ${\overset{ˇ}{R}}_{r}^{\left(j\right)}$, of the cluster specific covariance matrix estimate ${\overset{ˇ}{R}}^{\left(j\right)}$, is equal to,
(7)$$\begin{array}{ccc} {\overset{ˇ}{R}}_{r}^{\left(j\right)} & = & \sum_{e=1}^{Q_{j}}\sum_{j^{\prime }=1}^{N_{r}}{\left[{\tilde{Y}}_{r}^{\left(s,e,j\right)}\right]}_{.,j^{\prime }}{\left[{\tilde{Y}}_{r}^{\left(s,e,j\right)}\right]}_{.,j^{\prime }}^{T}\\ & & \;/\left(N_{r}Q_{j}-1\right)\;. \end{array}$$
Similarly, the imaginary part, ${\overset{ˇ}{R}}_{i}^{\left(j\right)}$, is equal to,
(8)$$\begin{array}{ccc} {\overset{ˇ}{R}}_{i}^{\left(j\right)} & = & \sum_{e=1}^{Q_{j}}\sum_{j^{\prime }=1}^{N_{r}}{\left[{\tilde{Y}}_{i}^{\left(s,e,j\right)}\right]}_{.,j^{\prime }}{\left[{\tilde{Y}}_{i}^{\left(s,e,j\right)}\right]}_{.,j^{\prime }}^{T}\\ & & \;/\left(N_{r}Q_{j}-1\right)\;. \end{array}$$
The cluster-conditional probability density function, Equation (Equation 4), is motivated by the asymptotic distribution of the tapered discrete-time Fourier transform of a multivariate, weak-sense stationary, random process (see, for example, Theorem 4.4.2, [54]), as well as the approximate independence of the multitaper eigencoefficients due to the orthogonality of the DPSSs [49]. The steps in the modified K-means algorithm are as follows:Specify the number of clusters.The initial observation labels are randomly guessed (as recommended in ref. [55]), and the cluster-specific parameters μ and R are estimated for each cluster using Equations (Equation 5), (Equation 7) and (Equation 8), to obtain $\overset{ˇ}{\mu }$, $\overset{ˇ}{R}$ for each cluster.Given $\overset{ˇ}{\mu },\overset{ˇ}{R}$ for each cluster, the feature matrices, $Y^{\left(s,e\right)}$, $e=1,\ldots ,N_{episode}$ (i.e., one for every 30-s episode), are re-clustered by assigning them the label that maximizes Equation (Equation 4).Using the new cluster assignments, $\overset{ˇ}{\mu }$ and $\overset{ˇ}{R}$ are re-computed.Steps 3 and 4 are repeated until an iteration occurs without a resulting change in the feature-matrix label. Once this occurs, subsequent iterations result in no change, and the algorithm has converged.

As previously stated, this iterative procedure is a variation on standard K-means clustering and is related to the K-means clustering algorithms discussed in [52,53,55]. For our data, by iteration 25, no changes to the eigencoefficient feature-matrix labels occur between one iteration and the next. The procedure consisting of steps 1 through to 5 above produces a single clustering. It remains to determine the number of clusters to use. Upon convergence, an approximate Akaike information criterion $A\tilde{I}C$ is computed. It is equal to,
(9)$$\begin{array}{c} A\tilde{I}C\left(N_{cluster}\right)=N_{cluster}\left(N_{c}^{2}+N_{c}\right)-\;\\ \;2\sum_{j=1}^{N_{cluster}}\sum_{e=1}^{N_{episode}}\ln P\left(Y^{\left(s,e\right)}\;|\;C=j\right)\;. \end{array}$$
with, as before, $N_{c}$ equal to the number of electrode contacts. As with the Akaike information criterion, for the purpose of model selection, models, or labelings, which minimize $A\tilde{I}C$ promote models that result in probable labeling and penalize models with many clusters. Smaller values of AIC imply a lower Kullback-Liebler divergence between the unknown data-generating probability density function and the modeled generating probability density function [56,57]. In this work, steps 1 through 5 above are computed, starting with $N_{cluster}$ equal to 2 through 14. The chosen clustering, ${\tilde{N}}_{contact}$, minimizes the $A\tilde{I}C$ specified in Equation (Equation 9). That is,
(10)$${\tilde{N}}_{contact}=\underset{n^{\prime }\in \left\{2,\ldots ,14\right\}}{arg min}A\tilde{I}C\left(n^{\prime }\right)\;.$$

### 2.8. REM Cluster

The goal of this work is to reliably identify a subset of REM episodes. This is accomplished once cluster labels have been assigned to the episode feature matrices, $Y^{\left(s,e\right)}$, $e=1,\ldots ,N_{episode}$, by identifying, in a sense to be made precise below, the cluster exhibiting the lowest alpha spectral power. Let $C_{j}$ be the set of episodes assigned to cluster *j*. A cluster-conditional multitaper estimate of alpha-power for contact *c* and subject *s* is equal to,
(11)$${\overset{ˇ}{S}}_{j}^{\left(c,s\right)}=\left.\sum_{e\in C_{j}}{\left.{\overset{ˇ}{S}}^{\left(c,s,e\right)}\left(f\right)\right|}_{f=10\;\mathrm{Hz}}/\left|C_{j}\right|\right.\;.$$
Next, for each contact, *c*, use ${\overset{ˇ}{S}}_{j}^{\left(c,s\right)}$, $j=1,\ldots ,N_{cluster}$, to rank the clusters from smallest ${\overset{ˇ}{S}}_{j}^{\left(c,s\right)}$ to largest. Specifically, let the rank $r_{j,c}^{\left(s\right)}$ equal,
(12)$$r_{j,c}^{\left(s\right)}=\underset{j^{\prime }\;=\;\left\{1,\ldots ,N_{cluster}\right\}}{\mathrm{rank}}{\overset{ˇ}{S}}_{j^{\prime }}^{\left(c,s\right)}\;.$$
The rank, $r_{j,c}^{\left(s\right)}$ is equal to 1 when ${\overset{ˇ}{S}}_{j}^{\left(c,s\right)}$ is the minimum of the set $\left\{{\overset{ˇ}{S}}_{1}^{\left(c,s\right)},\ldots ,{\overset{ˇ}{S}}_{N_{cluster}}^{\left(c,s\right)}\right\}$. Define the spectral rank, $\rho_{j}^{\left(s\right)}$ of cluster *j*, for subject *s*, to be the median of $r_{j,c}^{\left(s\right)}$ across electrode contacts:(13)$$\rho_{j}^{\left(s\right)}=\underset{c\;=\;\left\{1,\ldots ,N_{c}\right\}}{\mathrm{median}}r_{j,c}^{\left(s\right)}\;.$$
Finally, let the REM cluster index $j_{*}^{\left(s\right)}$ for subject *s* equal,
(14)$$j_{*}^{\left(s\right)}=\underset{j^{\prime }\;=\;\left\{1,\ldots ,N_{cluster}\right\}}{arg min}\rho_{j^{\prime }}^{\left(s\right)}\;.$$
The episodes belonging to $C_{j_{*}}$ are identified as REM sleep. If $j_{*}^{\left(s\right)}$ is integer. In the situation where $j_{*}$ is not integer-valued the analysis is deemed unreliable. For all data analyzed in this work, $j_{*}^{\left(s\right)}$ is integer.

### 2.9. Measures of Label Confidence

Two proxy measures of REM label identification specificity are computed. Let $j_{1,c}$ be equal to,
(15)$$j_{1,c}=\underset{j\in \left\{1,\ldots ,N_{cluster}\right\}}{arg min}{\overset{ˇ}{S}}_{j}^{\left(c,s\right)}\;,$$
and let $j_{2,c}$ equal,
(16)$$j_{2,c}=\underset{\begin{array}{c} j\in \left\{1,\ldots ,N_{cluster}\right\}\\ j\neq j_{1,c} \end{array}}{arg min}{\overset{ˇ}{S}}_{j}^{\left(c,s\right)}\;.$$
Then, the minimum, per-contact, alpha-power difference, $d\alpha_{c}$ is equal to,
(17)$$d\alpha_{c}={\overset{ˇ}{S}}_{j_{2,c}}^{\left(c,s\right)}-{\overset{ˇ}{S}}_{j_{1,c}}^{\left(c,s\right)}\;.$$
The first figure of merit, $FOM_{1}$, is equal to,
(18)$$FOM_{1}=\underset{c=1,\ldots ,N_{contact}}{arg min}d\alpha_{c}\;.$$
Smaller $FOM_{1}$ indicates that, for at least one electrode contact, the difference in alpha power between the two clusters exhibiting lowest alpha power is small. It is listed in row 4 of Table 3 and is least for the two worst-performing subjects. A second figure of merit, $FOM_{2}$, is equal to the fraction of the contacts, *c*, in cluster $j_{*}$ (i.e., the REM cluster), for which ${\overset{ˇ}{S}}_{j_{*}}^{\left(c,s\right)}$ is not minimum (see row 5, Table 3).

### 2.10. Hold-Out Analysis

The specificity of the proposed classifier depends on the sample size. Figure 2 shows the specificity of the proposed classifier (see Section 2.8) on random data splits. Data is randomly divided into two equally sized sets of data (manually identified REM and Wake episodes are each randomly split apart from each other and then combined), and the classifier is applied to each set. This random splitting and classification is repeated 50 times. The resulting REM identification specificity is plotted in Figure 2.

Dependence upon sample size is further quantified by repeating the above 50 random data splits, but in this case, changing the split to data divisions of varying size. The resulting boxplots of REM identification specificity are plotted in Figure 3.

### 2.11. Comparison with the Alpha Power Detector

The proposed unsupervised classifier introduced in Section 2.8 is an atypical classifier. It is insightful to compare this classifier with an alpha-power binary hypothesis test. To do this, the statistical power of the proposed classifier with the alpha-power test is matched by setting the latter test’s threshold such that the number of REM episodes identified by the two different methods is equal. The number of false positives resulting from classification using the two methods is compared in Figure 4. Of the subjects exhibiting normal sleep, the proposed classifier out-performs the equivalent statistical-powered hypothesis test based upon alpha power for five of eight subjects, and performs comparably on the remaining two subjects. For one subject, the specificity of the proposed classifier is 20% larger than the specificity demonstrated by the competing binary-hypothesis test.

### 2.12. Clustering Pseudo-Code

The following pseudo-code illustrates the classification of the REM and Wake sleep stages. Given a night of 30 s, multi-channel, data section,

Divide the data into 30 s, non-overlapping sections of data.Compute the $110N_{c}$ eiegencoefficients characterizing each 30 s section of data.Specify the number of clusters to be equal to 2.Randomly assign a cluster label to each of the 30 s sections of data.Compute each of the cluster statistics conditioned upon the cluster labels.For each 30 s section of data, and for each cluster, compute the log-probability of the observed section data conditioned upon each of the cluster labels.Assign to each 30 s section of data the cluster label that maximizes the log-probability of the observed 30 s section eigencoefficients.Repeat steps 4 through 7 until the cluster labels do not change from one repetition to the next.Compute the approximate AIC, $\tilde{AIC}$, for this number of clusters, using all of the log-probabilities.Repeat steps 3 through 9 but increment the number of clusters on each repetition until 14 clusters are used, or until an empty cluster results.

Steps 1 through 10 above are used to estimate the cluster labels, which are taken to be the labels associated with the smallest $\tilde{AIC}$.

### 2.13. REM Identification Pseudo-Code

To identify the REM cluster, first perform the clustering estimation described in Section 2.12. Then:Average the cluster-specific alpha spectral power across 30 s episodes.For each electrode contact, order the average cluster alpha spectral power across clusters.For each cluster, compute the median of the ranks across the contacts. This is the spectral rank.The REM cluster is the cluster with the smallest spectral rank.

### 2.14. Software Overview

The Pegasus Acquisition Software, Neuralynx, version 2.1.1 is used for data acquisition. Except for the spectrogram depicted in Figure 1, custom MATLAB scripts are used for all spectral analysis, clustering, signal processing, and statistical analysis. These scripts are available upon request. Multitaper spectral analysis dates back to the seminal paper published in 1982 [49]. Software available to perform multitaper analysis is available in the following languages: Fortran, R, C, Python, Julia, and MATLAB [34,58,59,60,61,62].

## 3. Results

Table 3 shows the results of applying the proposed classifier introduced in Section 2.8 to recorded iEEG data. The unsupervised, multitaper, alpha rhythm classifier introduced in Section 2.8 outperforms an equivalently powered alpha-power hypothesis test (see Figure 4 in Section 2.11) and attains specificity comparable to that obtained by experienced human sleep-scorers. It requires one-half of a night’s worth of REM and Wake episodes of healthy sleep to obtain results comparable to those obtained with a full night of data (see Figure 3 in Section 2.10). Since the time at which these episodes are recorded spans an entire night, for practical purposes, an entire night’s worth of data is required. The provided figure of merit, $FOM_{1}$, is lowest for the two worst-performing subjects (see row 3, Table 3), while $FOM_{2}$ does not correlate with performance (compare rows 2 and 4 of Table 3).

**Table 3 bioengineering-10-01009-t003:** The proposed method identifies REM in 6 out of 8 subjects exhibiting normal sleep with a specificity greater than or equal to 94 %. For the other two subjects, the specificity is 86% and 73%; respectively. It exhibits performance comparable to the inter-rater reliability exhibited by human sleep scorers [63].

Subject	1	2	3	4	5	6	8	9	7
Minutes of REM Correctly ID’ed	47.5	26.0	14.5	22.5	31.5	62.5	25.0	60.0	NA
Fraction of REM Labels Correct	0.98	0.95	1.00	0.73	0.98	1.00	0.86	0.94	0.00
Fraction of REM Correctly ID’ed	0.77	0.42	0.60	0.40	0.88	0.78	0.79	0.67	NA
Min. Alpha Power Diff. B/W Closest Competing Cluster and REM Cluster	0.79	0.07	0.09	$2.00\times 10^{-3}$	0.01	0.07	0.74	0.06	$6.00\times 10^{-3}$
Fraction of REM Cluster Electrode Alpha Power Not Minimum	0.00	0.10	0.20	0.08	0.09	0.00	0.00	0.18	0.00
Number of Clusters	14	12	10	14	8	6	14	14	14
Number of Electrode Contacts	12	9	10	12	11	12	3	11	2

## 4. Discussion

The proposed multitaper, multi-channel iEEG REM detector achieves a specificity greater than or equal to 94% on 6 out of 8 subjects exhibiting normal sleep. The figure-of-merit metric, $FOM_{1}$, reported in Table 3, and labeled “Minimum alpha power difference between the closest competing cluster with the REM identified cluster” differs by over an order of magnitude between the typical subject and the two worst performing subjects. This figure-of-merit responds to the separation in terms of alpha power between the closest cluster and the REM cluster itself. Subject selection based upon this criterion raises performance to a specificity of greater than or equal to 94% on 6 out of 7 subjects when identifying REM with the proposed method. Finally, it is worth noting that while the precision of the proposed detector is not high, it is sufficient to identify a median (minimum) minutes of REM equal to 28.75 (14.5) min per subject. This is identified from a single night of data when the typical hospital stay of a focal epilepsy patient is 1 to 2 weeks.

The proposed method is motivated by six main statistical signal processing considerations. The first (i) is to use sufficient information in the alpha frequency interval for the purpose of optimizing a binary detector (Wake vs. REM). Naively, this suggests the use of time-frequency distributions such as those employed in ref. [24] to classify sleep. It is important to note that there exists a bias/variance trade when temporally resolving frequency-domain structures. Resolution in time implies less temporal averaging. In many cases, this results in an increase in estimator variance, which in this case increases feature-domain variance, which can obscure clusters. That bias will move cluster centers; however, unless it moves them in an overlapping direction, this does not impact clustering performance for the purpose of detection. Finally, the potential information lost by not using a time-frequency distribution is the specific time within a 30 s episode at which a reproducible time-frequency event occurs. This information, if it exists, may further divide the REM cluster and necessitate an otherwise unnecessary cluster-merging step. Similarly, one might consider incorporating phase-phase and phase-amplitude spectral features (e.g., [51]). We seek to capture known average dynamics that occur in REM and, furthermore, the across-brain region timing of alpha oscillations. This suggests the use of multichannel multitaper spectral statistics [49,64], because (a) a multitaper spectrum estimator minimizes out-of-band bias while controlling the increase in variance associated with data tapering; (b) the multitaper eigencoefficients are concentrated within the frequency interval (f−W,f+W), are pairwise orthogonal on this interval, and form an approximate basis for this interval (and by symmetry for (−f−W,−f+W), for real-valued time-series). The second statistical signal processing consideration (ii) is to use the likelihood-ratio. It is known by the Neyman-Pearson lemma to provide the optimal binary detector (see, for example, ref. [65]). The third, fourth, and fifth considerations (iii,iv,v) are to allow for subject specific idiosyncracy (exhibited by ref. [41] and enhanced in iEEG due to variable epileptic focii), temporal changes of brain state, and corrupted data, such as that which results from recording during patient motion. These considerations suggest the use of unsupervised learning applied to the 30 s stretches of recording to be sleep-scored. Among the large number of unsupervised learning methods, K-means clustering is classical and is covered in books such as [66]. Finally (vi), there is a limited quantity of sleep-scored iEEG data from which to train a classifier. To generalize well, our detector must be simple and motivated by existing scientific and signal processing knowledge. These considerations lead us to select the minimum alpha power cluster as defined in Section 2.8 since Wake, motion artifacts, and other outliers all tend to increase the signal power observed in the alpha frequency interval.

The performance of the proposed method compared with that exhibited by a more simple alpha-power detector (see Figure 4, Section 2.11), suggests the use of the proposed detector. Specifically, the specificity is superior for 4 of 8 subjects, is the same for 1, slightly worse for 2, and exhibits a much greater specificity for one subject. Focusing on the impact of the performance improvement on this latter subject, should this performance improvement generalize, the proposed detector will avoid the expected 20% of incorrectly labeled REM episodes that would occur when using the simple alpha power detector on $1/8^{\mathrm{th}}$ of all iEEG data.

The bootstrap analysis (see Figure 2) indicates comparable detector performance across random data assignments for all but Subjects 4,8, and 7 (i.e., the worst-performing subjects). For subjectss exhibiting normal sleep, the detector is applied in a hold-out analysis. For these worse-performing subjects, performance on half a night of data is comparable to that attained from a night of data. See Figure 3 and Table 1. The bootstrap and hold-out analyses provide evidence that the specificity exhibited by the proposed multitaper detector will generalize.

This work can be extended by including the proposed REM detector as part of a more general sleep stage classifier. In this context, the sleep-stage transition probabilities may be used to potentially increase performance. The sample size of nine subjects is not large. It will be important to verify the performance of the proposed multitaper, multi-channel REM detector with larger, future iEEG data. Should the performance generalize, it provides a basis upon which REM can be identified in iEEG recordings unaccompanied by EOG and EMG. Identified REM activity can be correlated against, for example, the location of the epileptogenic zone for the purpose of improving the outcome of focal epilepsy surgery.

## 5. Conclusions

The proposed unsupervised multitaper classifier correctly identifies 36±6 min of REM in one night of recorded sleep (row 1, Table 3) while incorrectly labeling less than 10% of all labeled REM episodes for all but one subject; human rater reliability is estimated at near 80% [63]. This classifier outperforms the equivalent statistical-power classical test. Hold-out analysis indicates that when using one night’s worth of iEEG data, an accurate generalization of the method on new data is likely. The introduced multitaper multi-channel alpha-rhythm classifier further paves the way to making a large quantity of otherwise unusable iEEG data available for the purpose of studying sleep.

## Figures and Tables

**Figure 1 bioengineering-10-01009-f001:**
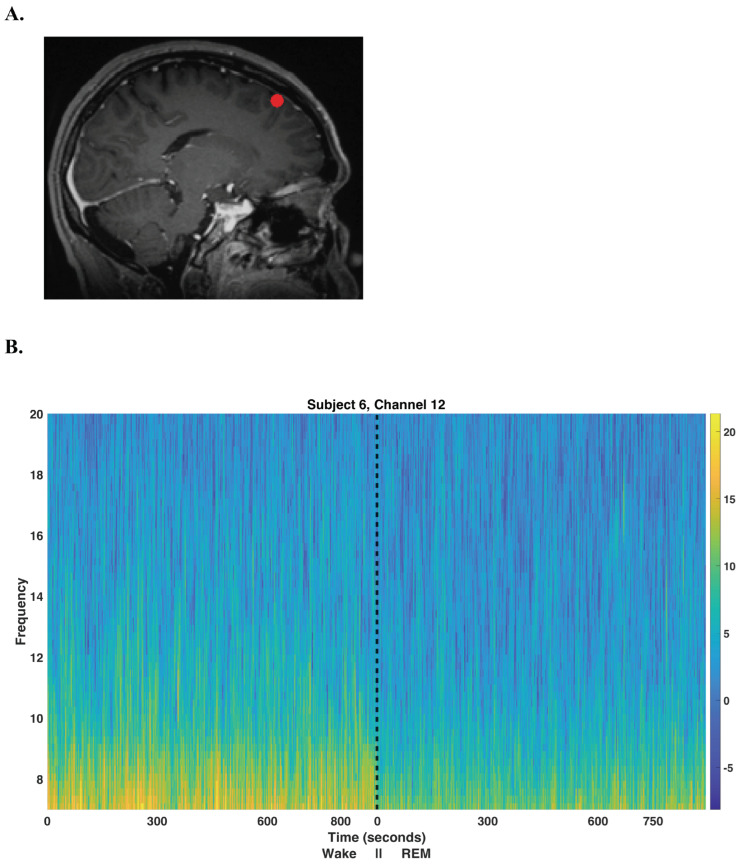
Electrode localization and spectrogram. (**A**) A sagittal MRI image showing the location of the outermost cortical contact of a frontal electrode in the left hemisphere for subject 6; the contact is located in the left superior frontal gyrus. (**B**) Spectrogram showing the alpha activity pattern detected from this electrode contact during Wake and REM sleep. The black line indicates the beginning of the REM sleep episode. Note that not shown is a large lapse in time-the REM episode occurred much later in the night than the wake period. The frequency (vertical axis) is in units of Hz. The multitaper spectrogram is computed using the Chronux MATLAB software, version 2.10 [34] using a 3 second duration sliding window with 50% overlap, a time-bandwidth parameter of 3 and 5 DPSS tapers.

**Figure 2 bioengineering-10-01009-f002:**
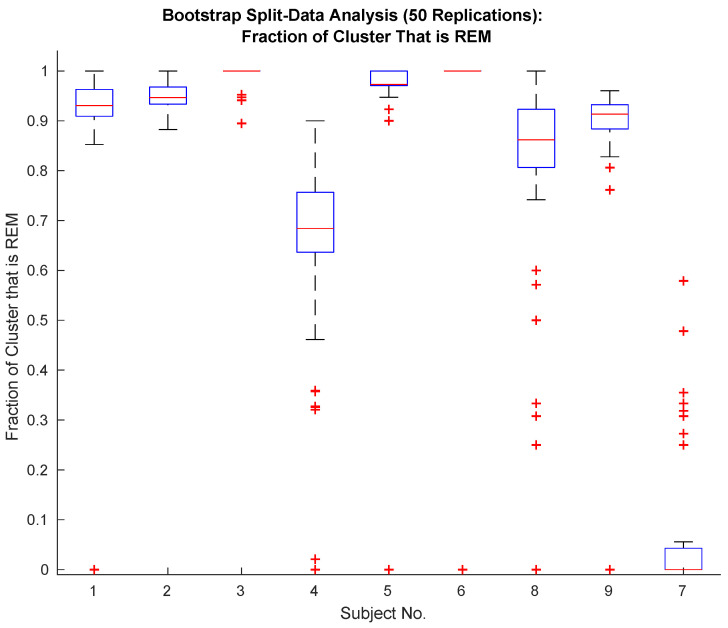
Performance of the proposed classifier is excellent with the exception of Subjects 4, 8 and 7. Subject 7 exhibits abnormal sleep and can be excluded along with Subject 4 based upon $FOM_{1}$ (see Table 3).

**Figure 3 bioengineering-10-01009-f003:**
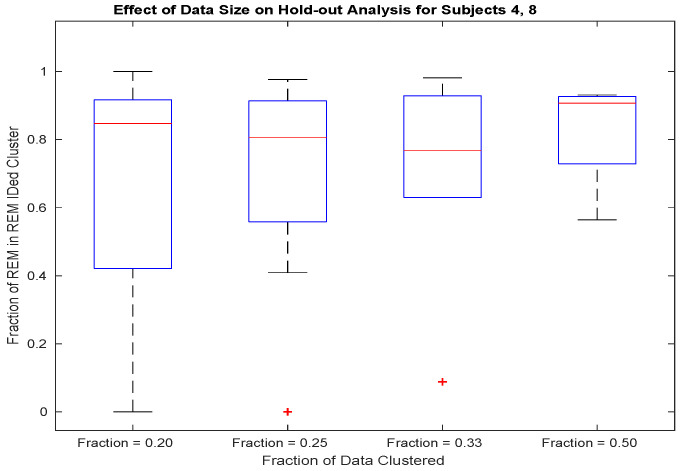
For subjects 4 and 8, to obtain specificity approaching full-data performance (see row 1 of Table 3), half of a night’s worth of REM and Wake episodes are required.

**Figure 4 bioengineering-10-01009-f004:**
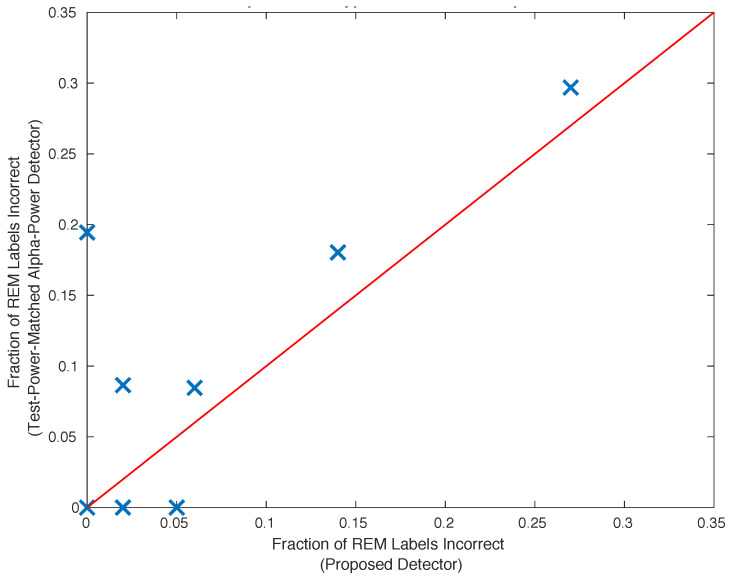
A matched-power comparison between the proposed classifier and a binary hypothesis test which compares alpha-power to reject a null hypothesis which is associated with REM sleep. The alpha-power threshold of the latter is set such that an equal number of REM episodes are identified by both classifiers. Of the subjects exhibiting normal sleep, the proposed classifier out-performs the equivalent statistical-powered hypothesis test based upon alpha power for six of eight subjects.

**Table 1 bioengineering-10-01009-t001:** Patient Information. The row for each patient provides demographic information, diagnostic information about their epilepsy, and brain imaging results.

Subject	Gender	Age	Handedness	Diagnosis	Imaging
1	F	24	R	Focal epilepsy, left and right temporal lobe	PET: Bilateral medial temporal hypometabolism. MRI: Normal
2	F	41	R	Focal epilepsy, left temporal lobe	PET: Borderline symmetric hypometabolism involving mesial temporal lobes. MRI: Normal
3	F	55	R	Medial refractory epilepsy with bilateral hippocampal foci	PET: Minimal hypometabolism of left medial and inferior temporal lobe. MRI: Normal
4	F	52	R	Intractable epilepsy, bilateral hippocampal	PET: Hypometabolism left para-hippocampal gyrus. MRI: Bilateral Hippocampal Cyst, 1-2 mm. Infarct, superior right cerebellum
5	M	29	R	Epilepsy of bilateral temporal origin.	PET: Hypometabolism bilaterally in the medial temporal lobes. MRI: Normal
6	M	19	L	Seizures of right temporal origin	PET: Hypometabolism in the right temporal lobe relative to left temporal lobe. MRI: Normal
7	F	20	R	Refractory epilepsy, left frontotemporal	PET: Small area of non-significant hypometabolism in left mesiotemporal region. MRI: Normal
8	M	38	R	Medically intractable epilepsy, left hemisphere	PET: Subtle hypometabolism predominantly in right parietal lobe. MRI: Normal
9	F	20	R	Medical refractory epilepsy with bilateral hippocampal foci	PET: Minimal hypometabolism of right lateral temporal lobe.MRI: Unremarkable (incidental finding of bilateral hippocampal cysts)

**Table 2 bioengineering-10-01009-t002:** Number and placement of implanted electrodes. Each electrode had multiple macro contacts, which are the actual recording surfaces located along the electrode shaft. Therefore, some macro contacts were located closer to the surface of the brain. Note that the macro contacts (6–10 per electrode) were evenly spaced along the electrode shaft.

Subject Number	No. of Electrodes Implanted	No. of Electrodes Implanted, by Location			
		**Left Hemisphere**		**Right Hemisphere**	
		**Frontal**	**Temporal**	**Frontal**	**Temporal**
1	16	4	4	4	4
2	16	4	4	4	4
3	16	4	4	4	4
4	12	3	3	3	3
5	14	3	4	3	4
6	11	3	2	3	3
7	2	0	2	0	0
8	4	0	2	0	2
9	12	3	3	3	3

## Data Availability

The data presented in this study are available on request from the corresponding author. The data are not publicly available due to privacy concerns and regulations.

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
