# Peer review of "Unsupervised Multitaper Spectral Method for Identifying REM Sleep in Intracranial EEG Recordings Lacking EOG/EMG Data"

_bioengineering, 2023, doi:10.3390/bioengineering10091009_

Round 1
Reviewer 1 Report
In this paper the authors propose a new unsupervised multitaper spectral method for identifying REM sleep in intracranial EEG recordings without EOG/EMG Data.
Overall the paper is well structured and clearly written. The proposed approach is interesting and seems to be useful but the authors must bring additional robustness to their claims. Please see the details below:
Abstract: “this extremely valuable data”, what data?
Introduction. The need for the proposed method must be better emphasized.
Figure 1. Axis has “700” only showing partially. Please include the spectrogram setup settings (window, overlap, FFT size, …)
Methodology: A good description of the tools that have been used would be interesting and useful for others that may try to replicate the work. Some pseudocode could facilitate understanding of the formulation sequence.
Were these tools used: DOI: 10.3791/60333 ?
Results: The authors mention: Proposed method is better when EOG/EMG is not available, then evidences must be shown; Proposed method is better than others in the same conditions, then a comparison with other methods must be presented. Additionally, the method showed to perform poorly in 3 of 9 cases, 33%, which is not a good result in clinical context. Authors should also enhance the description of the advantages of the proposed method.
It would be interesting to show results for other datasets or compare with other datasets.
Author Response
Thank you very much for your comments. Very helpful and appreciated.

Reviewer 2 Report
The paper presents Unsupervised Multitaper Spectral Method for Identifying REM Sleep using EEG signals. The major comments are
1. Introduction starts abruptly, which is not accepted at all. Authors are requested to add some solid background and explain the rationale behind the study.
2. The literature review is not done comprehensively. Authors should include relevant research to strengthen the literature. Some of the papers are listed below
Cascaded LSTM recurrent neural network for automated sleep stage classification using single-channel EEG signals - ScienceDirect
Multiclass sleep stage classification using artificial intelligence based time-frequency distribution and CNN - ScienceDirect
An Attention-Based Deep Learning Approach for Sleep Stage Classification With Single-Channel EEG | IEEE Journals & Magazine | IEEE Xplore
Application of data fusion for automated detection of children with developmental and mental disorders: A systematic review of the last decade - ScienceDirect
Entropy | Free Full-Text | Sleep Stage Classification Using EEG Signal Analysis: A Comprehensive Survey and New Investigation (mdpi.com)
Efficient sleep stage classification based on EEG signals | IEEE Conference Publication | IEEE Xplore
3. The authors are encouraged to identify the gaps and mention the contributions of the study.
4. The results are shallow. Please provide more results. In addition, the authors must also explain the reasons for their results.
5. The conclusion is again shallow- Please provide indepth analysis and conclusions. The current form is not acceptable at all.
6. Mention the limitations, merits, demerits, and future directions for the researchers.
Require proof-editing from local language expert.
Author Response
Thank you for your comments. They have been used to improve the quality of the manuscript.

Round 2
Reviewer 1 Report
The authors have correctly addressed all my comments. The paper has now publishing quality.
Reviewer 2 Report
The comments has been addressed. Paper can be accepted, congratulations.
NA